# Synergy of Small Antiviral Molecules on a Black-Phosphorus Nanocarrier: Machine Learning and Quantum Chemical Simulation Insights

**DOI:** 10.3390/molecules28083521

**Published:** 2023-04-17

**Authors:** Slimane Laref, Fouzi Harrou, Bin Wang, Ying Sun, Amel Laref, Taous-Meriem Laleg-Kirati, Takashi Gojobori, Xin Gao

**Affiliations:** 1Computational Bioscience Research Center (CBRC), King Abdullah University of Science & Technology (KAUST), Thuwal 23955-6900, Saudi Arabia; 2A Computer, Electrical and Mathematical Sciences and Engineering (CEMSE) Division, King Abdullah University of Science and Technology (KAUST), Thuwal 23955-6900, Saudi Arabia; 3Center for Interfacial Reaction Engineering (CIRE), School of Chemical, Biological and Materials Engineering, University of Oklahoma, Norman, OK 73019, USA; 4Department of Physics and Astronomy, College of Science, King Saud University, Riyadh 11451, Saudi Arabia

**Keywords:** MD, machine learning, DFT, inhibitor, black phosphorus, thermodynamic, molecular states, drug vehicles

## Abstract

Favipiravir (FP) and Ebselen (EB) belong to a broad range of antiviral drugs that have shown active potential as medications against many viruses. Employing molecular dynamics simulations and machine learning (ML) combined with van der Waals density functional theory, we have uncovered the binding characteristics of these two antiviral drugs on a phosphorene nanocarrier. Herein, by using four different machine learning models (i.e., Bagged Trees, Gaussian Process Regression (GPR), Support Vector Regression (SVR), and Regression Trees (RT)), the Hamiltonian and the interaction energy of antiviral molecules in a phosphorene monolayer are trained in an appropriate way. However, training efficient and accurate models for approximating the density functional theory (DFT) is the final step in using ML to aid in the design of new drugs. To improve the prediction accuracy, the Bayesian optimization approach has been employed to optimize the GPR, SVR, RT, and BT models. Results revealed that the GPR model obtained superior prediction performance with an $R^{2}$ of 0.9649, indicating that it can explain 96.49% of the data’s variability. Then, by means of DFT calculations, we examine the interaction characteristics and thermodynamic properties in a vacuum and a continuum solvent interface. These results illustrate that the hybrid drug is an enabled, functionalized 2D complex with vigorous thermostability. The change in Gibbs free energy at different surface charges and temperatures implies that the FP and EB molecules are allowed to adsorb from the gas phase onto the 2D monolayer at different pH conditions and high temperatures. The results reveal a valuable antiviral drug therapy loaded by 2D biomaterials that may possibly open a new way of auto-treating different diseases, such as SARS-CoV, in primary terms.

## 1. Introduction

The fast development of clinical biomedicine and nanobiotechnology has inspired the current generation of diverse inorganic nanoparticles that provide various pattern modalities as possible substitutions in addressing diseases by synergistic therapy, and principally to treat metastatic pathology [1,2,3]. Two-dimensional (2D) materials have fascinated extensive consideration due to their exceptional and distinctive properties [4,5,6]. Encouraged by the great success of graphene, a variety of 2D materials have been deployed for various property-related applications such as energy, environmental science, catalysis, physics, and biomedicine [7,8,9,10,11,12,13,14]. For example, transition metal dichalcogenides (TMDs), nitrides and carbonitrides (MXenes), and hexagonal boron nitride (h-BN), and their derivatives are realized with a monolayer and a few layers, which exhibit weak van der Waals interlayer bonding and strong in-plane covalent bonding [15,16,17]. Thereby, the biological behavior of Xenes 2D materials is under outstretched investigation to check their biosafety and biocompatibility, such as biodegradability, cytotoxicity, and toxic derivatives [18]. The broad spectra of potential uses and clinical implementation of the Xenes 2D systems in biomedical drug therapy still require proper control of biodegradability [19] and toxicity, as well as an explicit understanding of the biochemical procedures occurring in the drug molecules on 2D nanomaterials and their interfaces. The first aspect is particularly valuable since the understanding of the orientations of molecules with the multi-layered materials can characterize the process of raising biologically friendly materials for gene silencing, scattering complexes, hybrid pharmaceutic ingredients, and biomedicine sensing. In the present study, motivated by the latest developments in 2D nanomaterial synthesis and their encouraging solicitation in anti-tumor drugs [20,21,22], we conducted a comprehensive investigation in order to use 2D materials as nanocarriers for drug delivery. One such potential pharmacological option for treating various desists is ebselen (EB), while another is favipiravir (FP) [23,24]. Favipiravir has recently been used to treat infections caused by the Ebola virus, and it was found to be safe during clinical studies [25,26]. During the outbreaks of the SARS-CoV-1 and MERS-CoV infections, favipiravir was once more reviewed, and it was shown to be effective against these viruses in animal model systems [27,28,29,30,31]. Ebselen is an organoselenium compound with cytoprotective, anti-inflammatory, and anti-oxidant properties. This compound has previously been investigated to treat multiple diseases, including hearing loss [32] and bipolar syndrome [33]. Ebselen has minimal cytotoxicity [34], and its safety in humans has been estimated in a number of clinical trials [35,36,37,38]. Ebselen has the predominates inhibitor targeting the main protease (Mpro) of SARS-CoV-2 (e.g., Mopar is a crucial enzyme of coronaviruses) [24]. Recently, in vitro human cell studies reported that both favipiravir and ebselen could effectively stop the replication of SARS-CoV-2 at an earlier stage [24].

In recent years, the application of machine learning techniques in nanomaterials and pharmaceutical technologies has gained significant attention from researchers. Several studies have used backpropagated neural networks to model various systems [39,40], including Darcy–Forchheimer slip flow models for nanomaterials and ferrofluids [41], non-Fourier heat flux models for transient heat exchange [42], and ternary nanofluid flow between parallel plates [43]. The accuracy of these models was evaluated using reference datasets obtained via the Homotopy Analysis Method and numerical methods. The effects of non-dimensional parameters on the systems were also analyzed. In this paper, by utilizing accelerated molecular dynamics, machine learning, and quantum chemicals based on first-principles density functional theory (DFT), we briefly report the structural stability and thermochemical properties of FP and EB functionalized within a typical phosphorene 2D monolayer. The chemical bonding of FP and EB in 2D black-phosphorus materials is revealed by considering the vacuum and continuum solvent model. However, by calculating the Gibbs free energy changes of each step, we demonstrated that the temperature and pH values would be used to tune the releasing ratio of the antiviral molecules onto the phosphorene sheet. Our results provide new fundamental insights into exploring innovative 2D materials as new nanocarriers for antiviral drug therapy. These atomistic findings open the door to the parametrization of multiscale methodologies, such as contemporary machine learning based on electronic theories, continuum models, Monte Carlo simulations, and coarse-grained methods, to address the interplay between the interaction of drugs/2D monolayers with the cell membrane as a function of their morphology. In addition, we rigorously explored the efficiency of four machine learning models to accelerate the prediction of Hamiltonian and non-bond energy from EB, FP, and FP-EB, namely Bagged Trees, Gaussian Process Regression (GPR), Support Vector Regression (SVR), and Regression Trees (RT). The reason for considering multiple models is to compare and evaluate their performance in accurately predicting the behavior of the antiviral molecules in the 2D monolayer. We calibrated the machine learning models using Bayesian optimization based on the training data. Results revealed the superior performance of the GPR model by reaching an averaged $R^{2}$ of 0.9649. Training efficient and accurate models for approximating DFT is the final step in using machine learning to gain more insight into the design of new drugs.

The remainder of this paper is organized as follows. Section 2 presents the used data and the obtained results. In Section 3, we discuss the pharmachemical implications. Section 4 briefly describes the preliminary materials, including the Mean Squared Error (MSE), the investigated machine learning models, and ab initio calculations. Finally, we offer conclusions in Section 5.

## 2. Outcomes and Discussion

### 2.1. Data Description

Molecular dynamics is a powerful approach that enables atoms in the systems to interact with each other based on given parameters, such as temperature, time, pressure, and other conditions. The evolution of molecular simulations can be presented by various terms in the Hamiltonian (kinetic energy, potential energy, interaction energy, free energy). Thereby, the consistency of MD methods combined with ML and other quantum techniques can be considered. The main key in this work is to improve the Hamiltonian and non-bond energy terms from MD simulations as a starting point. The datasets are generated based on the COMPASS III [44] force field of molecular dynamics simulations for small antiviral molecules on BP single layer. By using the ML algorithms mentioned above, the Hamiltonian and the interaction energy are also trained at periodic boundary conditions (PBC) of polycyclic molecules on top of a phosphorene monolayer. Training efficient and accurate models for approximating DFT is the final step in using ML to aid in the design of new drugs.

### 2.2. Data Analysis

This part is dedicated to assessing the capability of the investigated machine learning models in predicting Hamiltonian and non-bond energy from three different datasets, MLFF-EB, MLFF-FP, and MLFF-FP_EB.

Figure 1 displays the distribution of the Hamiltonian energy from three different datasets. Plotting the probability density can be a useful tool for exploratory data analysis. Generally speaking, it can help visualize a data distribution and gain insights into its properties. It shows the relative frequency of different values and the shape of the distribution, which can identify patterns or anomalies. It can also be used to calculate statistical measures such as mean, variance, and skewness, providing further insight into the properties of the data. From Figure 1, we observe that these datasets are non-Gaussian-distributed. It would challenge traditional prediction methods, such as principal component regression, designed based on the Gaussian assumption of data. Thus, machine learning models designed without assumptions about the data distribution could be promising.

Figure 2a–c depicts the pairwise correlation coefficients between the multivariate data of antiviral drugs on the BP monolayer, MLFF-EB, MLFF-FP, and MLFF-FP_EB, respectively. As the calculation proceeds, there is no change in the temperature profile. This suggests that the model is dynamically stable. We observe that the temperature is absolutely correlated with kinetic energy. Once more, the temperature is uncorrelated with the other variables since it is related to the variations of the potential energy, non-bond energy, and total energy, respectively. Furthermore, it is also observed that in the Hamiltonian, potential energy, kinetic energy, non-bond energy, and total energy are constant throughout the simulation processes. We found a lowering in the kinetic energy that leads to general stability. However, this indicates that the model used in this study shows additional thermal stability of the system. The relaxed geometries are further used for creating a black phosphorus sheet for loading antiviral molecules that characterize the drug-BP based nanocarrier. Accordingly, the optimized ML structures (MLFF-EB, MLFF-FP, and MLFF-FP_EB, at T = 0 K) are in good agreement with the interaction energies and van der Waals energies found by means of DFT calculations [45]. For more information, please see Appendix A, respectively. Finally, during the calculations, the entire cell with its external pressure was fixed, and it had a moderate correlation with respect to bonding energy, angle energy, and TPE, as depicted in Figure 2.

### 2.3. Prediction Results

#### 2.3.1. Hamiltonian Energy Prediction

Three datasets were used to assess the performance of the machine learning models. Each sub-set of data consists of eight input variables (i.e., total kinetic energy, angle energy, torsion energy, inversion energy, van der Waals energy, non-bond energy, pressure, and temperature) and one out-variable (i.e., Hamiltonian energy). Each data set comprises around 2000 samples. In the first experiment, the dataset was divided into two subsets: training composed of 80% and testing with 20%. The k-fold cross-validation technique was considered in constructing these models based on the training data. Specifically, a 5-fold cross-validation procedure was employed in training the investigated models. The Bayesian optimization approach was employed to optimize the GPR, SVR, RT, and BT models to determine the optimal parameters that minimize the Mean Absolute Error between the actual and predicted Hamiltonian energy using the training data.

Table 1 summarizes the evaluation metrics, RMSE, MAE, $R^{2}$, and MAPE, computed from the testing datasets for MLFF-EB, MLFF-FP, and MLFF-FP_EB. Results revealed that the investigated machine learning models exhibited good capability in predicting Hamiltonian energy. We also notice from Table 1 that no single approach uniformly dominates the other models for the three considered datasets. For the MLFF-EB dataset, the SVR model attained the best prediction accuracy with an $R^{2}$ of 0.9767. In this case study, the GPR model obtained an accurate prediction with an $R^{2}$ of 0.9614 for the MLFF-EB dataset, and the best prediction quality was obtained by the GPR model with an $R^{2}$ of 0.9669 when applied to the MLFF-FP_EB dataset.

Table 2 lists the average evaluation metrics per model for Hamiltonian energy prediction. The closest $R^{2}$ value to 1 and the lowest RMSE, MAE, and MAPE values characterize the best prediction performance. Results in Table 2 show that all four models performed well in predicting the Hamiltonian energy, with $R^{2}$ values ranging from 0.9478 to 0.9649, indicating a good fit between the predicted and actual values. The GPR model had the highest $R^{2}$ value (0.9649) and the lowest values for RMSE, MAE, and MAPE, indicating that it performed slightly better than the other models in terms of accuracy. An $R^{2}$ value of 0.9649 means that the GPR model can capture 96.49% of the variation in the Hamiltonian energy of the antiviral molecules in the phosphorene monolayer. In addition, the GPR model achieved the lowest RMSE (0.0789) and MAE (0.0521) values among all models, indicating that the model had the smallest average deviation between predicted and actual values. Additionally, the GPR model had the lowest MAPE value (0.0095), indicating that the average percentage error between predicted and actual values was the lowest among all models. Therefore, the GPR model performed the best in terms of all evaluation metrics. This indicates that the model is highly accurate in predicting the Hamiltonian energy and can be used as a reliable tool for designing new antiviral molecules.

It is worth examining the prediction errors from the four investigated models (i.e., BT, GPR, SVR, and RT). The prediction error refers to the deviation between the true Hamiltonian energy measurements and their predicted values. Figure 3a–c displays the boxplots of the prediction errors for each model based on testing datasets from the MLFF-EB, MLFF-FP, and MLFF-FP_EB datasets, respectively. Visually, we observed that the prediction errors of the GPR and SVR models are concentrated around zero, indicating better prediction accuracy than the BT and RT models. Furthermore, it can also be seen that the GPR model with narrower boxes and whiskers reaches superior performance compared to the SVR for the MLFF-FP and MLFF-FP_EB datasets (Figure 3). Moreover, SVR and GPR models show relatively comparable performance under the MLFF-EB dataset (Figure 3).

As a visual illustration, the scatter graphs and plots of actual and predicted Hamiltonian energy of MLFF-EB, MLFF-FP, and MLFF-FP_EB, when applying the trained GPR model to testing data are depicted in Figure 4. To simplify visual readability, we presented only the results from the best model, and the results from the other models were omitted. We observe that the predicted values of Hamiltonian energy obtained by the GPR models are fairly close to the actual data of MLFF-EB, MLFF-FP, and MLFF-FP_EB, which indicates the promising performance of the GPR model.

#### 2.3.2. Non-Bond Energy Prediction

In the second experiment, the main key objective was to predict and improve the non-bond energy for MLFF-EB, MLFF-FP, and MLFF-FP_EB based on seven input variables (i.e., total kinetic energy, angle energy, torsion energy, inversion energy, van der Waals energy, pressure, and temperature) related to ab initio results [45]. Towards this end, we constructed four machine learning models (BT, SVR, GPR, and RT) based on a 5-fold cross-validation with the training dataset. The parameters of each model have been optimized using the Bayesian optimization approach in training. The trained models have been employed for non-bond energy based on testing datasets. Then, the statistical metrics ($R^{2}$, RMSE, MAE, and MAPE) are computed to compare the prediction performance of the investigated machine learning models (Table 3). For the three datasets, the GPR approach dominates the other approaches (i.e., BT, SVR, and RT) in terms of the four performance evaluation metrics (Table 3). Regarding the MLFF-FP, the four models provide relatively comparable performance with an $R^{2}$ around 0.97; the GPR model slightly outperformed the other models with an $R^{2}$ of 0.9792. Considering the MLFF-EB data, the GPR model obtained the best prediction performance for MLFF-EB and MLFF-FP_EB with an $R^{2}$ of 0.9735 and 0.9725, respectively. Table 4 lists the averaged evaluation metrics per model for non-bond energy prediction. The GPR model performed the best among all the models with an $R^{2}$ value of 0.9751, indicating a very good fit between the predicted and actual non-bond energies. The GPR model also had the lowest RMSE and MAE values, indicating that the predicted values were very close to the actual values. The MAPE value for the GPR model was 0.9579%, the lowest among all models, indicating that the model had the lowest percentage error in prediction. Therefore, based on these evaluation metrics, the GPR model was the best performer among all the models for non-bond energy prediction.

Figure 5a–c shows the boxplots of the prediction errors for the four models based on testing datasets (MLFF-EB, MLFF-FP, and MLFF-FP_EB). Visually, we observe that the boxplots are around zero, which means that the prediction quality is good and the prediction errors are small. Additionally, it can be seen that the GPR model provides relatively slightly less prediction errors than the other models for the three considered datasets.

The actual MD results and the predicted non-bond energy from the four models based on testing datasets are depicted in Figure 6a–c. We found that the predicted values of non-bond energy from the GPR model closely followed the measured data. Figure 6d–f shows the scatter plot of the measured and predicted values of non-bond energy from the GPR model. A scatter plot is a useful tool to visually compare a dataset’s predicted and measured data. In Figure 6d–f, the colored dots represent the data points, where the *x*-axis represents the predicted values and the *y*-axis represents the measured non-bond energy values. If the dots are closely clustered around the diagonal line, the model performs well and makes accurate predictions. Thus, from Figure 6d–f, we observe that the predictions are close to the measured values.

## 3. Pharmachemical Implications

### 3.1. Drug Release

#### 3.1.1. The pH Sensitivity

The 2D biomaterials are extensively applied in thermotherapy by tuning the drug–virus interactions. It is important to assign an effective nanocarrier for drug therapy; similarly, it is well-noted that the close link concerning virus and plasma antigen levels of tissue type (plasminogen) has a substantial clinical impact [46]. Accordingly, as a result of defection tissue, the normal and infected tissue should have different pH numbers. As a result, the stability of different respiratory diseases is the maximum at a pH close to 6, so the application of the voltages also affects the pH of the plasma tissue. The voltage induced is widely used in physical therapy to treat a variety of diseases. However, the pH measurement is simulated in the solvent environment by placing altered charged ions. On the other hand, the solid–solvent interface provides an alternative challenge for biomaterial simulation due to the large number of degrees of freedom hosted by the liquid states. For this purpose, we used an implicit solvation model, where the inhibitor molecule/2D biomaterial is treated at the level of a quantum mechanic state, and the solvent is treated as a continuum model described by the relative permittivity. This implicit solvation model is found to mimic the correct experimental solvation energy in various solvents [47].

To predict the pH sensitivity of the binding characteristics, we follow the method in Ref. [47] by adding or removing electrons number of valence states, expressed by the modification of the work function and the applied voltage, which has been demonstrated to provide consistent results with the experiments [48]. The adsorption energy of neutral FP, EB, and FP + EB on the BP single layer is −2.61, −5.41, and −8.86 meV/Å${}^{2}$, respectively; inclusion of 0.1 extra electrons will promote the energy of the system and cause a lowering of the binding energy. For example, adding half electrons in FP, EB, and FP + EB on BP surface changes the energy to 4.02, −1.95, and 2.82 meV/Å${}^{2}$, respectively, and the desorption of the hybrid FP + EB drugs becomes spontaneous. This scheme allows us to frequently control the chemical desorption by introducing fractional charge at the interface under the Poisson–Boltzmann approximation. In particular, pH-dependent adsorption depicted in Figure 7 suggests that larger adsorption energy is obtained in a polar environment with a higher pH value. Thus, when they are expected to release pharmaceutical drugs in a pointing position, an application of a low pH value with a smaller positive voltage would be appropriate.

#### 3.1.2. Thermotherapy Properties

The electronic and pH-influencing properties of antiviral drugs were described above. The key character of chemical adsorption nature related to temperature dependence is the next step in this investigation. Correspondingly, as represented in Figure 8, the temperature indeed alters the adsorption energy of the molecules at the BP monolayer, either at vacuum or solvent level. It can be well-noticed that by considering the vibrational correction, despite the fact that the entropy change is probably dominated by translational entropy rather than vibration, it shifts the adsorption to more positive or less negative numbers at a higher temperature. The zero-point energy at lower temperature values (0–100 K) has a minor effect on the energy (between 1–2 meV/Å${}^{2}$ for FP, EB, and FP + EB, respectively). On the other hand, increasing temperature from 150 to 420 K causes shifts of up to 4.1 meV/Å${}^{2}$ for FP, EB, and FP + EB on the BP-vacuum interface with the incorporation of the corrected vibrational free energy. Equivalent trends have been found by adding the continuum solvent level to these antiviral-molecules/BP 2D-system. Surprisingly, the adsorption remains stable up to 450 K, at the vacuum phase for hybrid FP + EB on the BP sheet. The thermotherapy is vigorous, and the releasing rate of the antiviral molecules will be highly enhanced due to the large drop in the adsorption energy at higher temperatures.

In general, the drug will have a larger release rate beyond 350 K, and it might be due to the height’s thermal deviation at a larger temperature. In the ex vivo evaluation process, the phosphorene-loaded drugs can be used as an anti-defected cell drug that has a pertinent act in the thermal therapeutic [45]. The temperature could upturn to around 450 K within a few minutes. The excellent therapeutic efficiency of these drugs in the black phosphorus nanocarrier is firmly associated to the large ability to target drug-release and the hyperthermia of the infected cell. Further, these findings suggest that the hybrid FP + EB therapy of low nanocarrier-based delivery systems can be activated effectively [48]. This comprehensive investigation will contribute to optimizing virus growth and storage conditions, facilitating the molecular characterization of this important pathogen. The interest of accommodating those inhibitors in close contact with the phosphorene sheet as a new delivery scheme is that *it could continuously control the desorption free energy* of the aromatic molecules by applying an electric field, together with the presence of external stimuli to tune their pH environment, and changing their temperature.

## 4. Materials and Methods

This section is dedicated to molecular dynamics simulations and machine learning models deployed for this study, which is related to describing nanocarrier loaded drugs prediction. ML approaches are effective methods for accelerating DFT relaxations and require significantly fewer iterations (DFT calculations) for convergence. It has shown excellent transferability along FP, EB, and hybrid FP and EB molecules physisorbed on a BP single layer, and closely describes the ab initio interaction energy of the entire system. This is an encouraging initiative toward a common class of force-field and ab initio approaches combined with ML for drug molecules on ultra-thin film crystalline 2D material.

### 4.1. Molecular Dynamics Simulations

Molecular dynamics (MD) simulations based on atomic force fields can accurately simulate the non-covalent geometry of a large number of molecular systems from non-equilibrium systems to thermodynamic states at the atomic level. In the present study, the current data were obtained by means of molecular dynamics simulations, as implemented in Forcite software [44]. To properly adsorb FP and EB drugs on a BP single layer, we used a (5 × 4) supercell model of black phosphorus containing 80 phosphorus atoms for the nanocarrier. A vacuum region of 20 Å is introduced perpendicular to the BP film to prevent unphysical periodic effects. A Hamiltonian including kinetic energy, potential energy, interaction energy, and free energy as a function of the change of temperature and pressure was simulated at 350 K and atmospheric pressure, respectively. We used an NVT canonical ensemble based on the Nosé thermostat. During the MD simulations, we adopted a time step of 0.3 fs for an ensemble of 100 ps of calculations to reach the thermodynamic states and accurate MD calculations.

### 4.2. Gaussian Process Regressor

GPR models, which are within nonparametric kernel-driven learning models, showed extended modeling ability for handling nonlinear prediction problems because of their nonlinear approximation capabilities [49,50,51]. They are well-known for their ability and flexibility in modeling small-sized data with a Gaussian or non-Gaussian distribution, as well as providing uncertainty measures on predictions [52,53]. Moreover, the GPR model’s benefit is its capacity to provide a confidence interval to evaluate the reliability of the predicted values, making it widely used in numerous applications [54]. The GPR models have been utilized in various applications, including renewable energy systems monitoring [50,55], COVID-19 spread prediction [56], and spatio-temporal PM${}_{2.5}$ prediction [57].

To introduce GPR, we have to consider the response y of a function f at the input x, which is expressed as [58],
(1)$$y_{i}=f\left(x_{i}\right)+\varepsilon_{i}.$$ Here, $\varepsilon ∼N(0,\sigma_{\varepsilon }^{2})$ is an additive noise, and *f*(***x***) is considered as a random variable. It is worth pointing out that the uncertainty on *f* could decrease significantly by the observation of the function’s output at different input points.

The function f(x) is considered following a Gaussian process. Accordingly, $y_{i}$ follows a joint Gaussian distribution [58]:(2)$$y={\left[y_{1},y_{2},\cdots \;y_{n}\right]}^{⊤}∼N\left(m\left(x\right),K+\sigma^{2}I\right),$$
where $m\left(x\right)={\left[m\left(x_{1}\right),m\left(x_{2}\right),\cdots \;m\left(x_{n}\right)\right]}^{⊤}$ refers to the vector of mean values m(·), I denotes the identity matrix, and K refers to the n×n covariance matrix with (i,j)th element $K_{ij}=k\left(x_{i},x_{j}\right)$. Within the GPR framework, $K_{ij}=k\left(x_{i},x_{j}\right)$ is termed the kernel function [52,58]. Numerous kernels have been used in the literature, including the Linear kernel, Rational Quadratic (RQ) kernel, Squared Exponential (SE) kernel, Matern 5/2 (M52) kernel, and Exponential (Exp) kernel. The two commonly used kernels (i.e., linear and squared exponential) are expressed respectively as [52]:(3)$$k_{\mathrm{Lin}}\left(x_{i},x_{j}\right)=\theta_{1}^{2}+\theta_{2}^{2}\left(x_{i}-\theta_{3}\right)\left(x_{j}-\theta_{3}\right).$$
(4)$$k_{\mathrm{SE}}\left(x_{i},x_{j}\right)=\theta_{1}\exp\left(\frac{{\left(x_{i}-x_{j}\right)}^{2}}{\theta_{2}}\right).$$

To optimize the GPR model in the training stage, we need to determine the kernel parameters that enable maximizing the following likelihood [58,59].
(5)$$\theta_{\mathrm{opt}}=\underset{\theta }{\arg\;max}L\left(\theta \right),$$
where $\theta =[\theta_{1},\theta_{2},\;\cdots ]$ are the parameters of the kernel (also called hyperparameters), the mean values m(.) are taken to be zero, and
(6)$$L\left(\theta \right)=\frac{1}{\sqrt{{\left(2\pi \right)}^{n}\;\left|K+\sigma^{2}I\right|}}\exp\;\left(-\frac{1}{2}\left(y^{⊤}\left(K+\sigma^{2}I\right)y\right)\right).$$

Many efforts have been reported in the literature to calibrate the hyper-parameters by using grid search or random search. Here, Bayesian optimization will be adopted to fine-tune the GPR hyperparameters by maximizing the marginal likelihood in (Equation 5) with respect to θ [60].

We assume $x_{*}$ is a new test input, then the estimated mean and variance associated with ${\hat{y}}_{*}=f\left(x_{*}\right)=f_{*}$ are [58,59]:(7)$${\hat{y}}_{*}=k_{*}^{⊤}{\left(K+\sigma^{2}I\right)}^{-1}y,$$
and
(8)$$\Sigma_{*}=k_{**}-k_{*}^{⊤}{\left(K+\sigma^{2}I\right)}^{-1}k_{*}.$$
respectively. Then, $y_{*}$ follows a conditional distribution which has the form:(9)$$y_{*}|y∼N\left({\hat{y}}_{*},\Sigma_{*}\right),$$
where K=k(X,X), $K_{**}=k\left(X_{*},X_{*}\right)$, and $K_{*}=k\left(X,X_{*}\right)$ are the covariance matrices computed based on the training data, the testing set, and both training and test sets, respectively.

Overall, GPR-based prediction involves two main steps. The first step is to estimate the hyperparameters, which include the kernel and noise variance, in the GPR model by maximizing the negative log marginalized likelihood based on the training data. The second step is to compute the predictive posterior distribution of GPR for new inputs, $X_{*}$, by using the mean of the predictive distribution in Equation (Equation 8) as the predicted value. For more details about the GPR model, see Refs. [49,50,58,61].

### 4.3. Bayesian Optimization Procedure

As discussed before, we need to calibrate hyperparameters during training to obtain good prediction performance when constructing a GPR model. Fine-tuning a machine learning model entails modifying its hyperparameters to optimize performance on a particular task. Optimal hyperparameter configuration has a direct impact on the model’s performance. Grid Search, Random Search, and Bayesian Optimization are popular methods for hyperparameter tuning in machine learning. Grid Search is a simple but computationally expensive method that exhaustively searches all possible combinations of hyperparameters within a predefined range. While Grid Search can guarantee to find the optimal combination of hyperparameters, it can be very time-consuming and computationally expensive, especially when the number of hyperparameters and their possible values are high [62]. Random Search is another method for hyperparameter tuning that randomly samples a set of hyperparameters from a predefined range. Unlike Grid Search, Random Search does not guarantee finding the optimal combination of hyperparameters, but it is generally faster and more computationally efficient than Grid Search [62]. On the other hand, Bayesian Optimization (BO) is a more advanced method that uses probabilistic models and previous evaluations to guide the Search [63]. It can quickly find good combinations of hyperparameters with fewer evaluations than Grid Search and Random Search.

This study adopts Bayesian optimization to find the optimal values of GPR’s hyperparameters [63]. Recently, the BO approach has been frequently employed to fine-tune hyperparameters in machine learning methods. This could be attributed to its design using Gaussian processes and Bayesian inference to reach global optimization [60]. Importantly, the BO procedure considers the previous evaluations to determine the hyperparameter set and then evaluates the next step, which enables reducing the time of the optimization procedure [64]. Moreover, the BO procedure can optimize functions without closed form [65]. The optimization process based on the BO procedure needs fewer iterations compared to a grid search optimization.

The main idea of the BO procedure consists of building a probabilistic proxy model for the objective function, employing prior experiments’ results as training data [66,67]. Essentially, the proxy model (e.g., the Gaussian process) is not expensive to calculate and enables getting pertinent information on where we should evaluate the true loss function to reach suitable results. Eventually, we consider now the problem of adjusting *m* hyperparameters $P=p_{1},\cdots ,p_{m}$. To this end, the aim is to determine
(10)$$P^{*}=\underset{P}{\arg\;min}g\left(P|{\left\{\left(x_{i},y_{i}\right)\right\}}_{i=1}^{n}\right),$$
where g denotes the the objective function [56]. The optimization procedure is controlled by an appropriate acquisition function that decides the next set of hyperparameters to be evaluated [68].

Figure 9 depicts the basic idea of the BO procedure to optimize the GPR model during the training phase. Specifically, the mean squared error (MSE) between the actual molecular dynamics data and the GPR predictions is computed at each iteration. The optimized model is obtained once the MSE converges to a small value, close to zero.

### 4.4. SVR Models

Here, we briefly introduce another widely used approach, the SVR model, a flexible data-based approach with good learning capacity via kernel tricks. The key idea underlying the SVR consists in mapping the train data to a higher dimensional space and conducting linear regression in that space. This SVR model is known to be efficient in dealing with nonlinear regression by using the kernel trick, which enables mapping the input features into high-dimensional feature spaces [69,70]. Moreover, the relevant concept used in designing the SVR model lies in structural risk minimization. It is demonstrated that SVR provides satisfactory performance with limited samples [71]. Thus, VVR models have been broadly exploited in numerous applications, such as solar irradiance prediction [72], wind power prediction [51], and anomaly detection [50]. This study will use optimized SVR via Bayesian optimization for comparison.

### 4.5. Bagged Tree Model

Likewise, we discuss another important machine learning model called the bagged tree (BT), also known as bootstrap aggregating [73]. The essence of this model is based on merging the benefit of the bagging technique and decision trees to enhance prediction quality. Importantly, in the BT model, bootstrap sampling is employed to generate numerous samples from the original dataset. Next, multiple distinct decision trees are built, and their outputs are aggregated to obtain the final output [74]. Hence, the prediction quality of the decision trees will be improved, the error will be reduced, and the overfitting issue in individual trees will be bypassed significantly [75,76].

### 4.6. Evaluation Metrics

In this study, we assess the accuracy of the predicting models using three metrics: root mean square error (RMSE), mean absolute error (MAE), mean absolute percentage error (MAPE), and $R^{2}$.
RMSE: Root Mean Squared Error measures the differences between predicted and actual values. It is calculated by taking the square root of the average of the squared differences between the predicted and actual values.
(11)$$RMSE=\sqrt{\frac{1}{n}\sum_{t=1}^{n}{\left(y_{t}-{\hat{y}}_{t}\right)}^{2}},$$MAPE: Mean Absolute Percentage Error is a measure of the accuracy of a model in predicting values. It is calculated as the average of the absolute percentage difference between predicted and actual values.
(12)$$MAPE=\frac{100}{n}\sum_{t=1}^{n}|\frac{y_{t}-{\hat{y}}_{t}}{y_{t}}|\%,$$MAE: Mean Absolute Error is a measure of the average magnitude of the errors in a set of predictions, without considering their direction. It is calculated as the average of the absolute differences between predicted and actual values.
(13)$$MAE=\frac{\sum_{t=1}^{n}\left|y_{t}-{\hat{y}}_{t}\right|}{n},$$R-squared: $R^{2}$ is a statistical measure that represents the proportion of the variance for a dependent variable that is explained by an independent variable or variables in a regression model. It is also known as the coefficient of determination, and its value ranges between 0 and 1.
(14)$$R^{2}=\frac{\sum_{t=1}^{n}{\left[\left(y_{t}-\bar{y}\right)\cdot \left({\hat{y}}_{t}-\bar{y}\right)\right]}^{2}}{\sqrt{\sum_{t=1}^{n}{\left(y_{t}-\bar{y}\right)}^{2}}\cdot \sqrt{\sum_{t=1}^{n}{\left({\hat{y}}_{t}-\bar{y}\right)}^{2}}},$$
 where $y_{t}$ is the number of Hamiltonian data, ${\hat{y}}_{t}$ is its corresponding forecasted Hamiltonian and non-bond energy, and *n* is the number of records. Importantly, a higher $R^{2}$ value indicates a better fit of the model to the data. For example, if the $R^{2}$ value is between 0.7 and 0.9, it is generally considered to indicate a good fit. An $R^{2}$ higher than 0.9 indicates a very good fit of the model to the data. It means that the model can explain a large proportion of the variability in the data. on the other hand, a lower MAPE value indicates better model prediction accuracy. Lower RMSE, MAE, and MAPE values would imply better precision and prediction quality. Generally, a MAPE value of less than 10% is considered good, while a value greater than 10% is considered poor. However, the threshold for an acceptable MAPE value can vary depending on the application and the specific domain. Therefore, it is important to consider the context and specific requirements when interpreting the MAPE value.

### 4.7. Prediction Framework

This work investigates the feasibility of machine learning methods to predict the Hamiltonian and non-bond energy terms from MD simulations. The prediction methodology employed in this study is depicted in Figure 10. We compared four machine learning methods, including BT, SVR, GPR, and RT. The prediction framework consists of two key steps: model construction and model prediction. The data were first preprocessed to a zero mean and unit variance by subtracting the mean, μ, and dividing by the standard deviation, σ.
(15)$$y_{s}=\left(y-\mu \right)/\sigma .$$

After obtaining the predicted values, we reversed this step. The normalized data were then split into training and testing sets. The training set was used to construct the machine-learning models. Here, we employ Bayesian optimization to determine the optimal parameters of the machine learning methods. We applied a five-fold cross-validation technique in training the investigated models. After that, the previously constructed models were then utilized to predict the Hamiltonian (or non-bond energy) in the testing step. The model’s accuracy was checked by comparing measured data to predicted data via the score indicators: $R^{2}$, RMSE, MAE, and MAPE. An $R^{2}$ closer to 1 and lower RMSE, MAE, and MAPE values reflect an accurate prediction.

### 4.8. Ab Initio Calculations

One of the most popular approaches in quantum chemistry is density functional theory (DFT) that computes a wide variety of properties of almost any kind of atomistic system, such as molecules, crystals, surfaces, interfaces, and even biomaterial carriers when combined with machine learning. Thus, as the next step of this study, all DFT calculations were carried out using the Vienna Ab Initio Simulation Package (VASP 5.4) [77,78]. All electrons with a projected augmented wave (PAW) formalism were used to model the electron–ion interactions [79]. The exchange–correlation functional contribution to the total energy was modeled using the generalized gradient approximation (GGA). In addition to the modified Perdew–Burke–Ernzerhof (PBE) [79], for treating the van der Waals (vdW), interaction corrections through the D3–BJ approach were incorporated in all calculations [80,81], as established by Grimme et al. [82,83]. We used a (5 × 4) supercell model of a black phosphorus monolayer containing 80 phosphorus atoms for the nanocarrier to adsorb FP and EB drugs. A vacuum region of 20 Å was introduced perpendicular to the BP film to prevent unphysical periodic effects. Given the large size of the supercell, numerical integration was employed over the Brillouin zone using a 3 × 3 × 1 Γ-centered k-point grid [84]. A denser 5 × 5 × 1 mesh was applied during the self-consistent field (SCF) for electronic and frequency calculations. The pseudo-wave functions were expanded in a plane-wave basis set with an energy cut-off of 450 eV. The accuracy was ensured by adopting energy convergence criteria of 1 × 10${}^{-5}$ eV. Therefore, structural relaxation was performed using the conjugate gradient technique until the force on each atom was below 0.01 eV/Å. This approach has been successfully employed in the past to describe fullerene adsorption on single-layer graphene [6].

## 5. Conclusions

We have systematically unveiled the interaction characteristics of inhibitor drugs on the surface of a phosphorene sheet by using accelerated molecular dynamics simulations, machine learning, and density functional theory calculations. Importantly, four machine learning models (i.e., BT, GPR, SVR, and RT) were employed to predict Hamiltonian and non-bond energy in EB, FP, and FP + EB. Results indicate the promising prediction performance of machine learning models. Specifically, the optimized GPR model obtained the best performance in this case study compared to the other investigated models. Specifically, the GPR model consistently outperformed the other three models (BT, SVR, and RT) for both Hamiltonian and Non-Bond energy predictions. In particular, the GPR model achieved the highest $R^{2}$ value of 0.9649 for Hamiltonian energy prediction and 0.9751 for Non-Bond energy prediction. Additionally, the GPR model had the lowest RMSE, MAE, and MAPE values for both energy predictions, indicating its high accuracy and precision in predicting the energy values. Therefore, the GPR model can be considered the best-performing model among the tested machine-learning models in this study. Further, regarding DFT calculations, we carried these out in a vacuum and an aqueous phase to evaluate the binding ability and thermochemical properties for active drug delivery. We found that the hybrid drugs are physisorbed and enable functionalization of the 2D phosphorene, and they show robust thermostability. All the drug molecules displayed strong van der Waals interactions when combined with the phosphorene sheet. Moreover, the calculated pH values and Gibbs free energy changes at different temperatures suggest that the FP and EB enabled the release of the drug from the 2D monolayer at high temperatures and different pH conditions. These findings would point to drug/2D biosystems as a new and potentially effective way to steer drug vehicle delivery in thermotherapy.

## Figures and Tables

**Figure 1 molecules-28-03521-f001:**
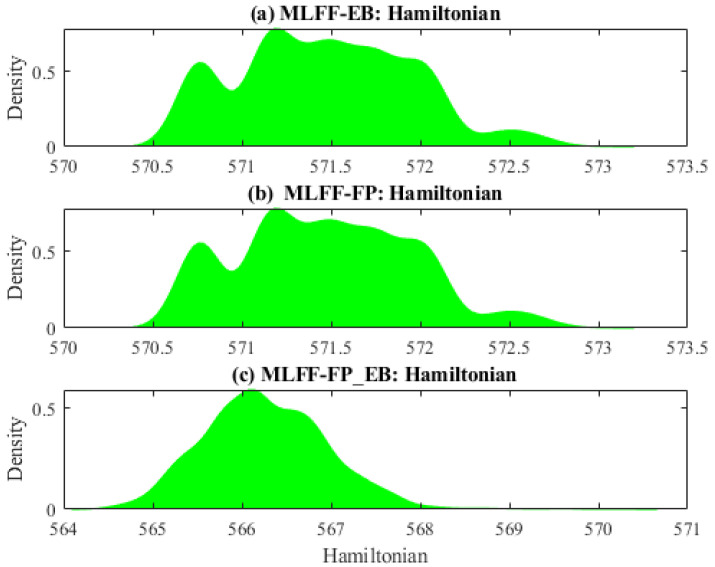
Distribution of Hamiltonian energy: (**a**) MLFF-EB, (**b**) MLFF-FP, and (**c**) MLFF-FP_EB.

**Figure 2 molecules-28-03521-f002:**
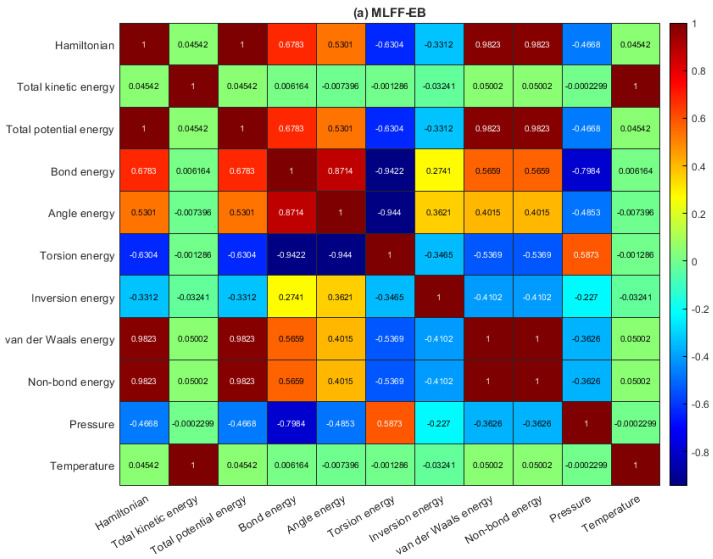
Heatmap of the correlation matrix: (**a**) MLFF-EB, (**b**) MLFF-FP, and (**c**) MLFF-FP_EB.

**Figure 3 molecules-28-03521-f003:**
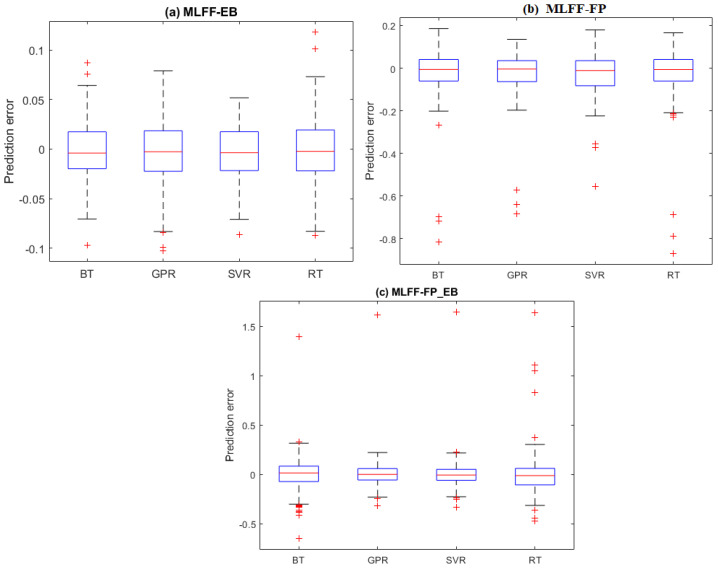
Prediction Distribution of prediction errors of the four models based on testing datasets: (**a**) MLFF-EB, (**b**) MLFF-FP, and (**c**) MLFF-FP_EB.

**Figure 4 molecules-28-03521-f004:**
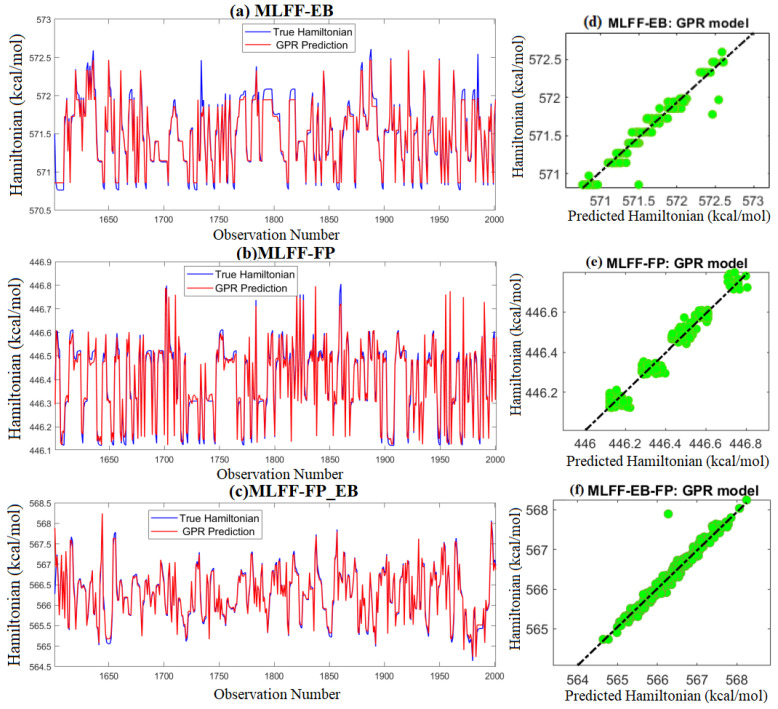
Predicted Hamiltonian energy from the GPR model based on test data: (**Top**) MLFF-EB, (**Middle**) MLFF-FP, and (**Bottom**) MLFF-FP_EB.

**Figure 5 molecules-28-03521-f005:**
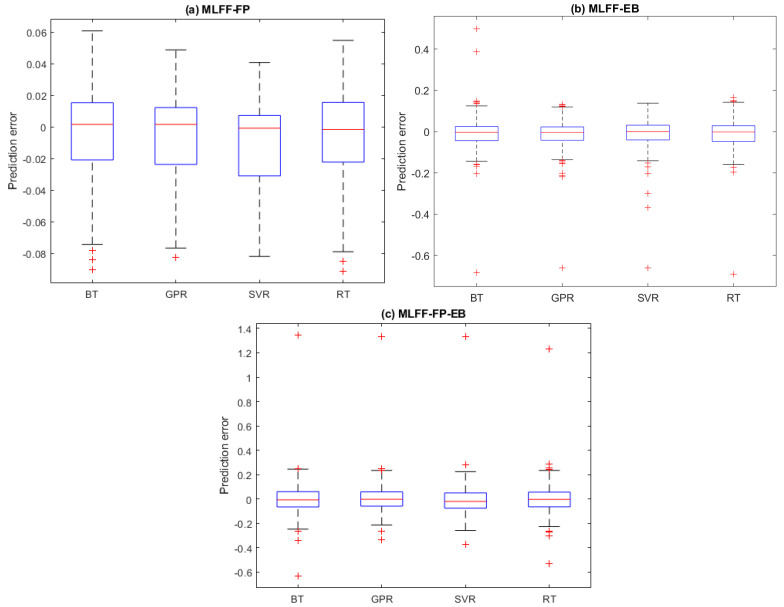
Prediction errors of the prediction models from the testing datasets: (**a**) MLFF-EB, (**b**) MLFF-FP, and (**c**) MLFF-FP_EB.

**Figure 6 molecules-28-03521-f006:**
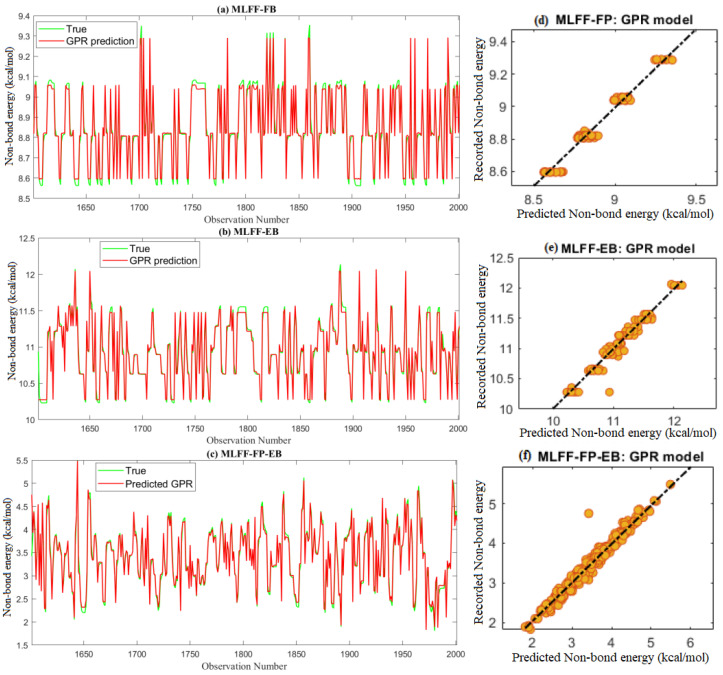
Predicted non-bond energy from the GPR model based on test data.

**Figure 7 molecules-28-03521-f007:**
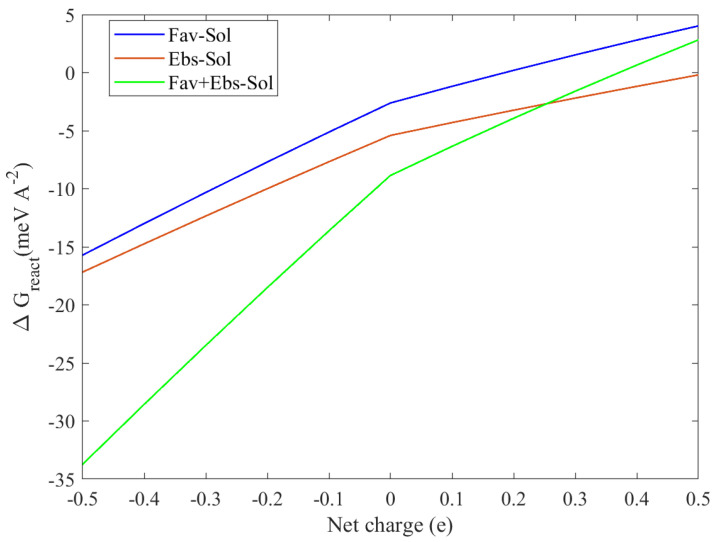
The calculated adsorption energy process as a function of net charge at the solvent phase of the BP sheet with favipiravir, ebselen, and favipiravir + ebselen hybrid drugs. The minus red and plus blue sign colors represent the difference between pH < 0 and pH > 0 values, respectively.

**Figure 8 molecules-28-03521-f008:**
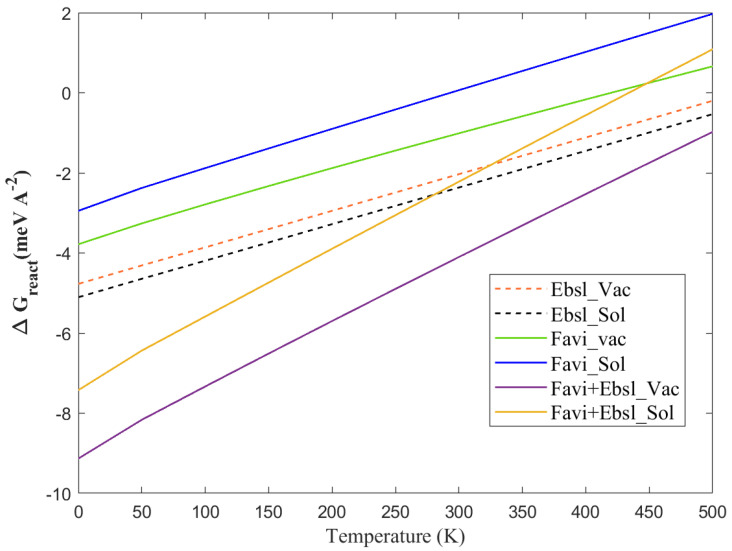
The free Gibbs adsorption energy process as a function of temperature at the vacuum and solvent phases of the BP sheet with favipiravir, ebselen, and favipiravir+ebselen hybrid drugs.

**Figure 9 molecules-28-03521-f009:**
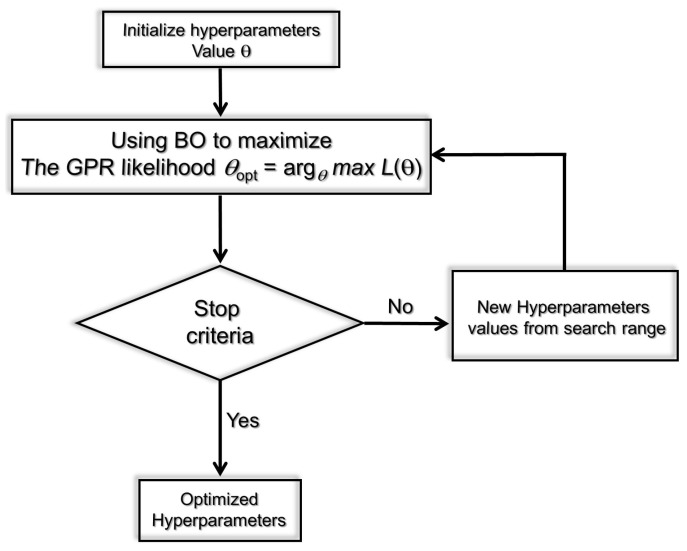
Schematic drawing of the main idea of the BO-based optimized GPR strategy.

**Figure 10 molecules-28-03521-f010:**
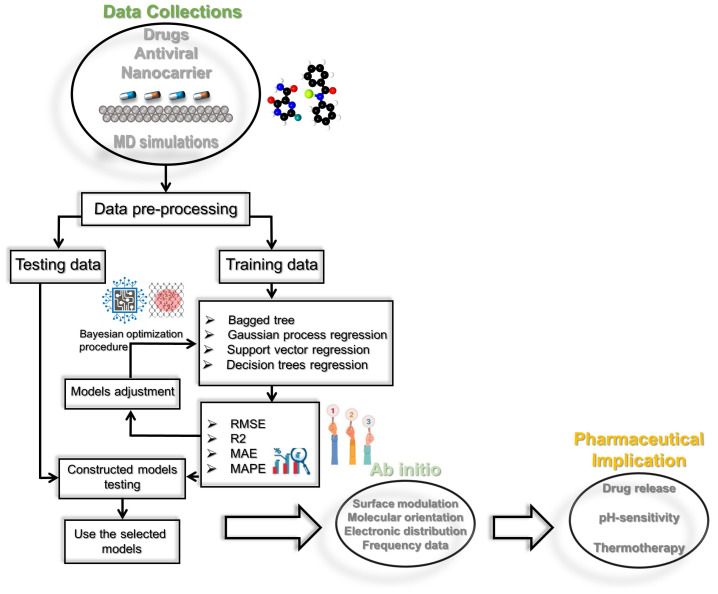
Illustration of the used prediction framework and general steps in antiviral drug discovery for small molecules on black phosphorus, developing quantitative structure–activity relationship models to optimize lead structures for pharmaceutical implication.

**Table 1 molecules-28-03521-t001:** Prediction quality of the Hamiltonian energy using the four optimized models.

		MLFF-FP		
Methods	$R^{2}$	RMSE	MAE	MAPE
BT	0.9743	0.0266	0.0217	0.0049
GPR	0.9663	0.0304	0.0238	0.0053
SVR	0.9767	0.0253	0.0208	0.0047
RT	0.9674	0.0299	0.0238	0.0053
		**MLFF-EB**		
Methods	$R^{2}$	RMSE	MAE	MAPE
BT	0.9513	0.1001	0.0655	0.0115
GPR	0.9614	0.0891	0.0606	0.0106
SVR	0.9447	0.1067	0.0809	0.0142
RT	0.9501	0.1014	0.0654	0.0114
		**MLFF-FP_EB**		
Methods	$R^{2}$	RMSE	MAE	MAPE
BT	0.9489	0.1459	0.1011	0.0179
GPR	0.9669	0.1173	0.0718	0.0127
SVR	0.9665	0.1182	0.0710	0.0125
RT	0.9259	0.1756	0.1109	0.0196

**Table 2 molecules-28-03521-t002:** Averaged evaluation metrics per model for Hamiltonian energy prediction.

	$R^{2}$	RMSE	MAE	MAPE
BT	0.9582	0.0909	0.0628	0.0114
GPR	0.9649	0.0789	0.0521	0.0095
SVR	0.9626	0.0834	0.0576	0.0105
RT	0.9478	0.1023	0.0667	0.0121

**Table 3 molecules-28-03521-t003:** Prediction quality of the non-bond energy.

		MLFF-FP		
	$R^{2}$	RMSE	MAE	MAPE
BT	0.9772	0.0279	0.0218	0.2466
GPR	0.9792	0.0266	0.0208	0.2348
SVR	0.9772	0.0279	0.0211	0.2382
RT	0.9775	0.0277	0.0221	0.2502
		**MLFF-EB**		
	$R^{2}$	RMSE	MAE	MAPE
BT	0.9685	0.0758	0.0493	0.4496
GPR	0.9735	0.0695	0.0468	0.4277
SVR	0.9705	0.0734	0.0492	0.4503
RT	0.9729	0.0703	0.0486	0.4448
		**MLFF-FP_EB**		
	$R^{2}$	RMSE	MAE	MAPE
BT	0.9648	0.1253	0.0843	2.5297
GPR	0.9725	0.1107	0.0732	2.2111
SVR	0.9710	0.1138	0.0771	2.3273
RT	0.9689	0.1178	0.0795	2.3512

**Table 4 molecules-28-03521-t004:** Averaged evaluation metrics per model for non-bond energy prediction.

	$R^{2}$	RMSE	MAE	MAPE
BT	0.9701	0.0763	0.0518	1.0753
GPR	0.9751	0.0689	0.0469	0.9579
SVR	0.9729	0.0717	0.0491	1.0053
RT	0.9731	0.0719	0.0501	1.0154

## Data Availability

The data and results associated with this article are available at https://github.com/SLIM23-CBRC/MD-ML-Hamiltonian (accessed on 1 April 2023).

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
