# Peer review of "Synergy of Small Antiviral Molecules on a Black-Phosphorus Nanocarrier: Machine Learning and Quantum Chemical Simulation Insights"

_molecules, 2023, doi:10.3390/molecules28083521_

Round 1

Reviewer 1 Report

The current investigation reports on the binding characteristics of Favipiravir (FP) and Ebselen (EB) antiviral drugs on a phosphorene nanocarrier using molecular dynamics simulations and machine learning (ML). The article is worthy to the scientific community after the following comments may be addressed/rebutted:

1.      The authors trained the data using ML approach with Bayesian optimization algorithm and concluded that “Results revealed that the GPR model obtained superior prediction performance with an R2 of 0.9649”. Usually the value of R2 will be one or closer to one.  Here it is 0.9649. I suggest the authors to train the data again to get R2 to be closer to one.

2.      Do you fin any other optimization algorithm better than Bayesian optimization algorithm?

3.      Include a shot note on density functional theory?

4.      What’s the significance of Gibbs free energy in the ongoing research work?

5.      The significance, characteristics, applications and recent developments in the field of machine learning should be included in the introduction section. Further, the improvement of introduction is required by adding the following studies based on machine learning

a)      Neural artificial networking for nonlinear Darcy–Forchheimer nanofluidic slip flow

b)      Intelligent backpropagated neural networks application on Darcy-Forchheimer ferrofluid slip flow system

c)      Backpropagated Neural Network Modeling for the Non-Fourier Thermal Analysis of a Moving Plate

d)      Numerical and Levenberg–Marquardt backpropagation neural networks computation of ternary nanofluid flow across parallel plates with Nield boundary conditions

e)      Intelligent neuro-computing for entropy generated Darcy–Forchheimer​ mixed convective fluid flow

6.      Why the author considered four different machine learning models. Mention the advantages of these methods over traditional machine learning models

7.      What are the significant factors that influence the success rate and efficiency of machine learning models

8.      How did you choose the damping parameter in optimization algorithm

9.      What scheme is used to update the damping parameter in optimization algorithm

Author Response

Reply to the Editor and Reviewers' comments

Title: Synergy of Small Antiviral Molecules on a Black–Phosphorus Nanocarrier: Machine Learning and Quantum Chemical Simulation Insights

Manuscript number: Molecules-2309552

Journal: Molecules (ISSN 1420-3049)

April 3, 2023

The authors would like to thank the editor for handling and reporting the reviewer comments of this paper. We also gratefully thank all reviewers for their efforts and the time they spent to review this paper and we really appreciate their valuable comments, suggestions and questions that helped us greatly in improving the quality of this manuscript as well as in clarifying several points in the earlier version of our paper. Both editor and reviewers' suggestions have been wisely incorporated in the revised version of the manuscript. Changes made are highlighted in red in the manuscript. We hope that our revision has improved the paper to a level of the reviewers’ satisfaction. Please find below our replies to the reviewers' comments (written in blue) and description of the changes made.

Response to Reviewer #1 comments:  The current investigation reports on the binding characteristics of Favipiravir (FP) and Ebselen (EB) antiviral drugs on a phosphorene nanocarrier using molecular dynamics simulations and machine learning (ML). The article is worthy to the scientific community after the following comments may be addressed/rebutted:

-      Comment 1: The authors trained the data using ML approach with Bayesian optimization algorithm and concluded that “Results revealed that the GPR model obtained superior prediction performance with an R2 of 0.9649”. Usually the value of R2 will be one or closer to one.  Here it is 0.9649. I suggest the authors to train the data again to get R2 to be closer to one.

Authors' answer: R2, also known as the coefficient of determination, is a statistical measure representing the proportion of the variability in the dependent variable explained by the independent variables in a regression model. R2 is often used as a measure of the goodness of fit of a regression model, and a higher R2 value generally indicates a better fit. In other words, R2 measures how well the regression model fits the data. It is a value between 0 and 1, where 0 indicates that the model does not explain any of the variability in the dependent variable, and 1 indicates that the model perfectly explains all of the variability in the dependent variable.

Results revealed that the GPR model obtained superior prediction performance with an R2 of 0.9649. Please note that the R2 value of 0.9649 indicates that the GPR model was able to explain 96.49% of the variability observed in the data, which is considered to be a very good prediction performance.

The R2 value alone may not be sufficient to evaluate the performance of the GPR model fully. In this study, three other metrics, root mean squared error (RMSE), mean absolute error (MAE), and mean absolute percentage error (MAPE), are also used in this study to evaluate the accuracy of the model's predictions. Prediction results in terms of RMSE, MAE, and MAPE confirm the good performance obtained by the GPR model. We explained that in the revised manuscript.

-      Comment 2:  Do you find any other optimization algorithm better than Bayesian optimization algorithm?

Authors' answer:  There are several methods for hyperparameter tuning in machine learning, including Grid Search, Random Search, and Bayesian Optimization.

Grid Search is a simple but computationally expensive method that exhaustively searches all possible combinations of hyperparameters within a predefined range. While Grid Search can guarantee to find the optimal combination of hyperparameters, it can be very time-consuming and computationally expensive, especially when the number of hyperparameters and their possible values are high.

Random Search is another method for hyperparameter tuning that randomly samples a set of hyperparameters from a predefined range. Unlike Grid Search, Random Search does not guarantee to find the optimal combination of hyperparameters, but it is generally faster and more computationally efficient than Grid Search.

Bayesian Optimization, on the other hand, is a more advanced method that uses probabilistic models and previous evaluations to guide the Search. It can quickly find good combinations of hyperparameters with fewer evaluations than Grid Search and Random Search. Bayesian Optimization is particularly useful for optimizing complex models with a large number of hyperparameters, where other methods can be computationally infeasible.

We clarified that in the revised manuscript (please see 1st paragraph in Section 2.3).

Comment 3: Include a shot note on density functional theory?

Authors' answer: Thank you for the valuable feedback. We have include a paragraph concerning a density functional theory right after the methodology.

Comment 4: What’s the significance of Gibbs free energy in the ongoing research work?

Authors' answer: Fundamentally, sine this is a chemical reaction the practical importance of the Gibbs energy is to make predictions based on the properties (ΔG values) of the reactants and products themselves, and reduces the need of experiment mechanisms in the lab that needs too much time and big effort. Although, the free energy is whether used to determine how systems change and how much antiviral molecules can produce in contact with biomaterial surface. So, it is expressed in two forms: the Helmholtz free energy F, sometimes called the work function, and the Gibbs free energy ΔG. Additionally, ΔG it enables to predict the direction of spontaneous change for a system under the constraints of constant temperature and pressure. These constraints generally apply to all living organisms. The significance of the change in Gibbs free energy for a chemical reaction is to help in determining the direction of the spontaneity of the reaction. In simpler terms, it explains whether the spontaneity of the reaction will be forward, backward, or equilibrium.

Comment 5: The significance, characteristics, applications and recent developments in the field of machine learning should be included in the introduction section. Further, the improvement of introduction is required by adding the following studies based on machine learning

  1. a) Neural artificial networking for nonlinear Darcy–Forchheimer nanofluidic slip flow
  2. b) Intelligent backpropagated neural networks application on Darcy-Forchheimer ferrofluid slip flow system
  3. c) Backpropagated Neural Network Modeling for the Non-Fourier Thermal Analysis of a Moving Plate
  4. d) Numerical and Levenberg–Marquardt backpropagation neural networks computation of ternary nanofluid flow across parallel plates with Nield boundary conditions
  5. e) Intelligent neuro-computing for entropy generated Darcy–Forchheimer mixed convective fluid flow

Authors' answer:  Thank you for sharing with us these interesting papers. As recommended, we updated the introduced by adding the suggested studies.

Comment 6: Why the author considered four different machine-learning models. Mention the advantages of these methods over traditional machine learning models

Authors' answer: Thank you for the valuable comment. This study considered four different machine-learning models, Bagged Trees, Gaussian Process Regression (GPR), Support Vector Regression (SVR), and Regression Trees (RT), to predict the Hamiltonian and interaction energy of the antiviral molecules in the phosphorene monolayer. The reason for considering multiple models is to compare and evaluate their performance in predicting the behavior of the antiviral molecules in the 2D monolayer accurately. This approach ensures that the final results are robust and reliable.

The advantages of these methods over traditional machine learning models include:

  1. Improved prediction accuracy: The use of advanced machine learning techniques, such as Bagged Trees, GPR, SVR, and RT can improve prediction accuracy and reduce the margin of error in the results.
  2. Ability to handle complex data: Traditional machine learning models can struggle to handle complex data sets with high dimensionality, but advanced techniques, such as GPR and SVR are capable of handling these types of data sets.
  3. Reduced risk of overfitting: Advanced machine learning models, such as bagged trees and regression trees, are less prone to overfitting than traditional models. This reduces the risk of models learning the noise in the data rather than the underlying patterns.

Overall, the use of advanced machine learning models can help to improve the accuracy and reliability of the results, especially in complex systems like the interaction of antiviral drugs on a 2D monolayer.

Comment 7: What are the significant factors that influence the success rate and efficiency of machine learning models

Authors' answer:  Generally speaking, several factors can significantly influence the success rate and efficiency of machine learning models, including:

  1. Quality and Quantity of data: Machine learning models require large, high-quality datasets to achieve high levels of accuracy. Poor quality data or small sample sizes can lead to overfitting, making the model less reliable.
  2. Hyperparameters tuning: Hyperparameters are parameters that are set before training a model, such as learning rate or regularization strength. Tuning these hyperparameters can significantly improve model performance and accuracy.
  3. Feature selection: Feature selection is the process of selecting the most relevant variables or features to include in the model. Including irrelevant or redundant features can negatively impact model performance, while omitting critical features can result in a lack of accuracy.

Comment 8: How did you choose the damping parameter in optimization algorithm

Authors' answer:  Thank you for the valuable comment. Here, Bayesian optimization is utilized to search for optimal hyperparameter values. The Mean Squared Error (MSE) is used as the objective function to evaluate the difference between the predicted values and the actual measurements. The training and hyperparameter search are stopped once the minimum MSE is achieved, indicating the best possible fit of the model to the data.

Comment 9: What scheme is used to update the damping parameter in optimization algorithm

Authors' answer:  Bayesian optimization is a sequential model-based optimization technique that involves constructing a probabilistic model of the objective function and using it to choose the next set of hyperparameters to evaluate. The damping parameter in Bayesian optimization refers to a hyperparameter that controls the degree of exploration versus exploitation in the search for the optimum. It is typically used in conjunction with the acquisition function, which is used to select the next set of hyperparameters to evaluate based on the current model of the objective function. The damping parameter can be tuned to balance the trade-off between exploring new regions of the search space and exploiting regions that have already been explored. A higher damping parameter encourages more exploration, while a lower damping parameter encourages more exploitation.

We have addressed all the comments above and we are very grateful to the reviewers for their valuable and helpful suggestions. Once again thank you for taking the time to share your constructive feedback for the purpose of improving our manuscript.

Reviewer 2 Report

Thank you for your efforts.

The main objective of this paper is the use of machine learning (ML) models to aid in the design of new drugs.

The topic is relevant in the field.

The added value is the use of ML models.

The conclusions consistent with the evidence and arguments presented.

I have some comments that must be considered in the revised version.

-------------------------------------------------------

1) The (Introduction) is too long including a very large number of references that are suitable for a paper (Not a thesis).

2) The mathematical model is difficult to follow. Can you summarize the steps in a paragraph?

3) In Fig. 3, illustrate what do you mean by "density" and what is its acceptable range?

4) Table 2 is very clear for different metrics. It will be more suitable to add a row for any previous work for comparison, or at least to show the acceptable levels of these metrics.

5) Same last note (4) in Tables 3 and 4.

6) In the graphs of Fig. 8, is the straight line the ideal case and the colored symbols are the practical ones (your findings). Please clarify.

7) The (Conclusion) is preferred to have some (main) numerical values of the (main) findings). I do not prefer to put a figure in the (Conclusion). I think Fig. 10 is put here by mistake.

8) References are up to date.... a good point (you have one in 2023, four in 2022 and 8 in 2021).

Author Response

Reply to the Editor and Reviewers' comments

Title: Synergy of Small Antiviral Molecules on a Black–Phosphorus Nanocarrier: Machine Learning and Quantum Chemical Simulation Insights

Manuscript number: Molecules-2309552

Journal: Molecules (ISSN 1420-3049)

April 3, 2023

The authors would like to thank the editor for handling and reporting the reviewer comments of this paper. We also gratefully thank all reviewers for their efforts and the time they spent to review this paper and we really appreciate their valuable comments, suggestions and questions that helped us greatly in improving the quality of this manuscript as well as in clarifying several points in the earlier version of our paper. Both editor and reviewers' suggestions have been wisely incorporated in the revised version of the manuscript. Changes made are highlighted in red in the manuscript. We hope that our revision has improved the paper to a level of the reviewers’ satisfaction. Please find below our replies to the reviewers' comments (written in blue) and description of the changes made.

Response to Reviewer #2 comments:

Thank you for your efforts.  The main objective of this paper is the use of machine learning (ML) models to aid in the design of new drugs.  The topic is relevant in the field.  The added value is the use of ML models.  The conclusions consistent with the evidence and arguments presented.  I have some comments that must be considered in the revised version.

Authors' answer:  We greatly appreciate the reviewer’s efforts to carefully review the paper and the valuable suggestions offered. We feel that the suggested changes have improved the standard of the paper significantly.

-      Comment 1: The (Introduction) is too long including a very large number of references that are suitable for a paper (Not a thesis).

Authors' answer: We thank the reviewer for pointing out this comment. To show the objective of this study and be more explicit we have reduced the main introduction to one and half-short pages. Please see the introduction in Section 1 of the revised manuscript.

-      Comment 2: The mathematical model is difficult to follow. Can you summarize the steps in a paragraph?

Authors' answer: Thank you for the valuable feedback. As recommended the main steps of the model has been summarized in a paragraph, please the last paragraph in Section 2.2 of the revised manuscript.

-      Comment 3: In Fig. 3, illustrate what do you mean by "density" and what is its acceptable range?

Authors' answer: Thank you for the feedback. By plotting the probability density function of the data, we can visualize its distribution and gain insights into its properties, such as its mean and variance. Plotting the probability density can be a useful tool for exploratory data analysis and can help inform subsequent analyses or modeling efforts. Specifically, the probability density function shows the relative frequency of different values in the data set and the shape of the distribution. This can help to identify patterns or anomalies in the data, which can be useful for understanding the underlying phenomena or making predictions. Additionally, the probability density function can be used to calculate statistical measures such as mean, variance, and skewness, which can provide further insight into the properties of the data.

For example, Fig. 3 shows that the distribution of Hamiltonian and non-bond energy is non-Gaussian, making it challenging to use traditional prediction methods, such as principal component regression, which are based on the Gaussian assumption of data. However, machine learning models, such as the GPR model used in this study, designed without assumptions about the data distribution, could be promising.

We explained that in the revised manuscript; please see the second paragraph in Section 3.2.

-      Comment 4: Table 2 is very clear for different metrics. It will be more suitable to add a row for any previous work for comparison, or at least to show the acceptable levels of these metrics.

Authors' answer: Thank you for your professional comment. In the absence of other works using the same dataset, we cannot include comparison with SOTA methods. The definitions for the four metrics used in the study have been included in Section 2.6 of the updated manuscript. As recommended, we included the acceptable levels if the evaluation metrics in the revised manuscript. In addition, in the paper, the boxplot and scatter plot analysis provide a visual confirmation of the model's reliability and performance. The compactness of the prediction errors around zero shows that the models can capture the variability of the data effectively. The scatter plot further demonstrates the closeness of the predicted observations to the actual observations, indicating that the models are able to make accurate predictions. This type of analysis is useful in evaluating the performance of machine learning models and can provide valuable insights into their reliability and effectiveness.

-      Comment 4: Table 2 is very clear for different metrics. It will be more suitable to add a row for any previous work for comparison, or at least to show the acceptable levels of these metrics.

-      Comment 5: Same last note (4) in Tables 3 and 4.

Authors' answer: Thank you for the valuable comment.

The four evaluation metric used in this study (i.e., R2, RMSE, MAE, and MAPE) are commonly used to assess the prediction performance of machine learning models. R2, also known as the coefficient of determination, measures the proportion of the variance in the dependent variable that is predictable from the independent variable. A higher R2 value indicates a better fit of the model to the data. For example, if the R2 value is between 0.7 and 0.9, it is generally considered to indicate a good fit.  In our case,  an R2 value of 0.96 indicates a very strong correlation between the predicted values and the actual values. This means that the model is able to explain 96% of the variance in the data, which is considered very good.

MAPE, which stands for Mean Absolute Percentage Error, measures the average percentage difference between the predicted values and the actual values. Generally, MAPE values below 10% are considered good, values between 10-20% are acceptable, and values above 20% are poor. However, the acceptable level of MAPE can vary depending on the specific application and industry standards. Here, MAPE values ranging from 0.25% to 2.5% indicate a very good prediction performance of the machine learning models used in the study. This means that, on average, the models' predictions deviate from the actual values by only a small percentage, which suggests that they are accurately capturing the underlying patterns in the data.

RMSE, which stands for Root Mean Squared Error, measures the average difference between the predicted values and the actual values. MAE, which stands for Mean Absolute Error, measures the average absolute difference between the predicted values and the actual values. Similarly, for RMSE, and MAE lower values indicate better performance.

We clarified that in the revised manuscript.

-      Comment 6: In the graphs of Fig. 8, is the straight line the ideal case and the colored symbols are the practical ones (your findings). Please clarify.

Authors' answer:  Thank you for the feedback. In the left panels of Fig. 8, we presented scatter plots of the recorded non-bond energy vs. the predicted using the GPR model.  A scatter plot is a useful tool to visually compare the predicted and recorded values of a dataset.  In the graph, we use colored dots to represent the data points, where the x-axis represents the predicted values and the y-axis represents the recorded values. If the predicted values are close to the actual values, the dots will be close to the diagonal line representing perfect prediction. By examining the scatter plot, one can easily see the relationship between the predicted and actual values. If the dots are closely clustered around the diagonal line, it indicates that the model is performing well and making accurate predictions. However, if the dots are scattered widely and not closely clustered around the diagonal line, it suggests that the model is not making accurate predictions. We clarified that in the revised manuscript.

-      Comment 7: The (Conclusion) is preferred to have some (main) numerical values of the (main) findings). I do not prefer to put a figure in the (Conclusion). I think Fig. 10 is put here by mistake.

Authors' answer: Thank you for pointing this out. We agree with this comment. As recommended, we quantitative results of the main findings in the conclusion section of the revised manuscript. Also, we updated the placement of Fig. 10 in the text.

-      Comment 8: References are up to date.... a good point (you have one in 2023, four in 2022 and 8 in 2021).

Authors' answer: Thank you for your professional and positive feedback.

We have addressed all the comments above and we are very grateful to the reviewers for their valuable and helpful suggestions. Once again thank you for taking the time to share your constructive feedback for the purpose of improving our manuscript.

Round 2

Reviewer 1 Report

The authors have addressed all my questions and concerns. Paper can be accepted in present form.